# OpenReview forum: "Automatic Chain of Thought Prompting in Large Language Models"
_ICLR.cc/2023/Conference — ICLR 2023 poster_

### Official Review · Reviewer_psXT · 2022-10-24

**Confidence:** 5
**Correctness:** 4
**Technical Novelty And Significance:** 2
**Empirical Novelty And Significance:** 2
**Recommendation:** 3

**Clarity, Quality, Novelty And Reproducibility:**

The paper is very clearly written and easy to follow. All the results can be reproduced well.

**Strength And Weaknesses:**

### Strengths:
- The paper proposes to counter the problem of manually designing prompts for various tasks by first clustering the samples (test) into different clusters using sentenceBERT, finding the most representative from each cluster, and using the Zero-shot-CoT approach on them to design prompts. This method is better than randomly sampling n-samples from the dataset and using Zero-shot-CoT prompts over them.
- The results of ten tasks demonstrate the effectiveness of the method with very competitive results on each one of them.

### Weaknesses:
- Firstly, when mentioning that the method automatically finds prompts using chain of thoughts (Auto_CoT) approach, this does not imply finding the best samples from the dataset and using the same method as before. In other words, the proposed method does not find the best chain of thought prompts as claimed but rather believes that “let's do this step by step” is the best zero-shot-CoT prompting strategy and the method merely uses that with different prompt samples. The title and the abstract are highly misleading in that way.
- Secondly, this is very obvious to anyone that the prompts must be representative of the test samples (as much as possible). This work takes the dataset, clusters them, finds the most representative sample from each cluster, and uses that as prompts. There is nothing extra-ordinary about this approach and is merely a small improvement over the randomly sampling strategy. This can also be seen from a marginal improvement in the results over a random sampling strategy.
- The paper suggests that the manual prompt search will be reduced with this approach but rather introduces a clustering strategy, and a fixed hyper-parameter might or might not be apt for each dataset and requires manual intervention. If the number of clusters go beyond the number of prompts a large LM can handle, I am not sure what can be done in those situations.
- Finally, if clustering was the way to go, the authors could have tried to cluster samples into n-clusters, see where the test sample is fitting, and could have tried to use just one prompt instead of k different ones at each time. This might have reduced the compute cost and would be interesting to see the performance vs resource trade-off.


**Summary Of The Paper:**

Prompting has been an important strategy lately to query large language models (LLMs). Chain of thought (CoT) prompting has demonstrated state-of-the-art on many datasets by forcing the models to generate intermediate steps while solving a problem. Some other manual designs of prompts (for example, appending “let's solve this step by step” to the solution steps) have improved the reasoning capabilities of LLMs on various tasks leading to superior results than normal prompting. However, designing prompts is a manual task often requiring lots of brute force to find a suitable prompt. In this work, the authors propose a strategy to automatically design a chain of prompting prompts known as Auto-CoT. On public benchmarks, Auto-CoT performs competitively to the manual CoT prompts.

**Summary Of The Review:**

The paper discusses a way to automatically create prompts which is a very promising direction. However, the work does not target how to create the prompts, rather uses different samples with the previously proposed prompting strategy. The approach used by the authors to cluster the samples together and construct prompts using the most representative samples improves the results marginally, but the bottleneck is not the samples for prompts but how the prompts are constructed. The work does not target that and is just an obvious extension of the other prompting strategies of the past. I would recommend then authors to extend their approach to find suitable prompts and not just suitable samples which is not that effective.

---

> ### Author Response · Authors · 2022-11-16
> **Response to Reviewer psXT**
>
> Thanks for your insightful comments. We humbly think that some concerns are caused by misunderstanding, which we will explain in detail below. We hope that our response can clarify the misunderstandings and you can consider our work more favorably.
>
> > 1. "Firstly, when mentioning that the method automatically finds prompts using chain of thoughts (Auto-CoT) approach, this does not imply finding the best samples from the dataset and using the same method as before. In other words, the proposed method does not find the best chain of thought prompts as claimed but rather believes that 'let's do this step by step' is the best zero-shot-CoT prompting strategy and the method merely uses that with different prompt samples. The title and the abstract are highly misleading in that way."
>
> ```Automatic``` means automatically constructing few-shot demonstrations as opposed to the manual designs in Manual-CoT. In other words, Auto-CoT is a substantial departure from the current paradigm of Manual-CoT. Manual-CoT applies in-context learning with few-shot demonstrations and has dominated the performance in CoT studies. However, the performance relies on manually designing both questions and reasoning chains. To eliminate such manual designs, we propose Auto-CoT, which constructs those few-shot demonstrations by automatically sampling questions via clustering and generating reasoning chains via invoking Zero-Shot-CoT. Thus, (i) ```automatic``` refers to automatically sampling diverse questions and automatically generating reasoning chains; (ii) ```automatic``` lies in the scope of few-shot demonstrations instead of finding the best (Zero-Shot-CoT) prompts. In contrast to Manual-CoT, Zero-Shot-CoT can be seen as an easily replaceable tool in our framework if a better prompt is available. We have added more clarity in the abstract.
>
> As with any new paradigms, it is expected that there is still space for improvements, e.g., finding better Zero-Shot-CoT prompts than ```Let's think step by step```. Given that API access is often hidden behind the paywalls of LLM providers (e.g., GPT-3), optimizing the prompts requires much more computation costs. In [1], ```Let's think step by step``` has already been shown as the current best prompt for Zero-Shot-CoT after comprehensive studies on 16 different prompts. Nonetheless, we have also conducted further experiments to modify the prompts with three different ones and see if the prompt can be optimized (Table 17). We find that optimizing the zero-shot prompts does not result in substantial benefits. We have added the discussion in Appendix D.4.
>
> > 2. "Secondly, this is very obvious to anyone that the prompts must be representative of the test samples (as much as possible). This work takes the dataset, clusters them, finds the most representative sample from each cluster, and uses that as prompts. There is nothing extra-ordinary about this approach and is merely a small improvement over the randomly sampling strategy. This can also be seen from a marginal improvement in the results over a random sampling strategy."
>
> In the research line of chain-of-thought prompting for LLMs, our method is the first step to constructing CoT prompts without needing the hand-craft of effective few-shot demonstrations like in Manual-CoT. To better compare the results with the random sampling strategy, we have conducted further experiments of Random-Q-CoT on all ten datasets in Table 3. As our goal is to eliminate the human input in Manual-CoT, Random-Q-CoT is obviously incompetent with inferior performance. In contrast, Auto-CoT (i) performs significantly better than Random-Q-CoT  in all the ten datasets at the significance level $p < 0.05$ (or even $p < 0.01$ for $6$ out of $10$); (ii) matches or even exceeds the performance of Manual-CoT.
>
> The superior performance of Auto-CoT is owing to our careful designs, including (i) a clustering-based method to obtain the most representative (diverse) questions and (ii) simple heuristics to reduce the risk of sampling wrong reasoning chains. We have further performed ablation studies in Section 5.3 to justify these two design components (with more comprehensive results in Appendix C.2).

---

> > ### Author Response · Authors · 2022-11-16
> > **Response to Reviewer psXT (continued)**
> >
> >
> > > 3. "The paper suggests that the manual prompt search will be reduced with this approach but rather introduces a clustering strategy, and a fixed hyper-parameter might or might not be apt for each dataset and requires manual intervention. If the number of clusters go beyond the number of prompts a large LM can handle, I am not sure what can be done in those situations."
> >
> > Auto-CoT requires no manual intervention in our experiments. The only hyper-parameter is the number of clusters, which equals to the number of desired demonstrations. This value is a result of extensive analysis in prior works [2]. Therefore, we re-use this value for both effectiveness and fair comparisons. Besides, using 8 demonstrations is far below what LLMs can handle (e.g., up to 2,048 tokens in GPT-3 models). Under the current settings, like prior studies, our method does not suffer from your mentioned situation.
> >
> > Nonetheless, we have performed further experiments to investigate the effect of the aforementioned hyper-parameter (number of clusters: $k$). Specifically, we set $k$ to varying values and illustrate the results in Figure 14. Our findings agree with [2] on the number of desired demonstrations ( = number of clusters: $k$).
> > Therefore, it is an effective default value. We have added the discussion in Appendix C.5.
> >
> > > 4. "Finally, if clustering was the way to go, the authors could have tried to cluster samples into $n$-clusters, see where the test sample is fitting, and could have tried to use just one prompt instead of $k$ different ones at each time. This might have reduced the compute cost and would be interesting to see the performance vs resource trade-off."
> >
> > This is a very smart idea. Following your advice, we have conducted experiments to evaluate its effectiveness (Figure 16). More concretely, we first cluster samples into $n$ clusters ($n = 2, \ldots, 10$) and then use just one demonstration from the cluster where the test sample fits. We observe that the performance of this method (denotes as "One-Shot") is on par with Random-Q-CoT that leverages $n$ demonstrations but is inferior to Auto-CoT. This method reduces the computation cost and can be seen as a less effective yet efficient version of Auto-CoT. We have added the discussion in Appendix D.5.
> >
> > **References:**
> >
> > [1] Takeshi Kojima, Shixiang Shane Gu, Machel Reid, Yutaka Matsuo, and Yusuke Iwasawa. Large
> > language models are zero-shot reasoners. In Thirty-sixth Conference on Neural Information
> > Processing Systems (NeurIPS), 2022.
> >
> > [2] Jason Wei, Xuezhi Wang, Dale Schuurmans, Maarten Bosma, Ed Chi, Quoc Le, and Denny Zhou. Chain of thought prompting elicits reasoning in large language models. In Thirty-sixth Conference on Neural Information Processing Systems (NeurIPS), 2022.

---

> ### Author Response · Authors · 2022-11-29
> **Kind Reminder**
>
> Dear Reviewer,
>
> We provided our response two weeks ago. Does it address your questions? We are more than happy to answer any further questions.
>
> Thanks!

---

> ### Author Response · Authors · 2022-12-04
> **Author Rebuttal Acknowledgement**
>
> Dear Reviewer,
>
> Thanks again for your efforts in reviewing our paper. Would you mind acknowledging our rebuttal? As the discussion due is approaching, if you have any questions, let's discuss on the OpenReview system.
>
> Best,
> Authors

---

### Official Review · Reviewer_G2Cc · 2022-10-25

**Confidence:** 4
**Clarity, Quality, Novelty And Reproducibility:** Clear enough to me.
**Correctness:** 3
**Technical Novelty And Significance:** 3
**Empirical Novelty And Significance:** 4
**Recommendation:** 8

**Strength And Weaknesses:**

# Strengths

- This is a fairly straightforward paper that attempts to (broadly) combine strengths of both manual chain-of-thought prompting with zero-shot chain of thought prompting. The idea is (mostly) simple and is likely to be of wide interest to the community.
- Fairly diverse set of results across arithmetic/commonsense/symbolic reasoning, with two LLMs are explored, GPT-3 and Codex, both via the OpenAI API.
-  I like the didactic/exploratory nature of section 3. I like that the paper reads like an investigation, and begins with a failure case, as well as diagnosing reasons for the failure, before then presenting the real method. This feels more organic and the lessons learned along the way are insightful (though I think the analysis is not as straightforward as it could be - see weaknesses)

# Weaknesses

I have one primary hesitation with the paper that prevents me from assigning it a higher score, as well as some secondary (less major) concerns.

## Primary: Lack of understanding of why Auto CoT works

As I mentioned, I like the section 3 analysis which analyzes exactly why fancy retrieval (vs random retrieval) works. But it seems like for Auto CoT, a lot of this nuance is lost, and we don't have a clear understanding of why this method is doing better, especially given that there are some ad-hoc decisions being made here.

If I understand correctly, Auto CoT is similar to Random-Q-CoT except (1) we enforce sampling from diverse clusters of questions and (2) heuristics to filter for simple rationales. Both of these modifications are presented at once in Table 3, without ablation studies, and compared only to Manual CoT. Strangely the Section 3 methods are omitted from this table (Random-Q-CoT, Retrieval-Q-CoT), but I think their inclusion would add a ton of insight into what exactly is causing gains for Auto CoT.

Again I think it is quite important to have some controls/ablations that explain why Auto CoT improves. The baselines I imagine are:

1. How well does the simple random retrieval + zero-shot CoT, i.e. Random-Q-CoT, do in Table 3? Again I think it's strange that this baseline is omitted here.
2. How much are the simple heuristics buying you? I am concerned that the simple heuristics are entirely responsible for the improved performance across tasks (i.e. it's just easier to learn from simpler demonstrations). Appendix C.2 is useful for understanding how heuristics improve Auto CoT but it's done while keeping the kmeans clustering fixed. It also does not measure the ultimate metric we care about, which is downstream task performance (instead focusing on "number of mistakes" in sampled demonstrations) I would love Auto CoT run without kmeans, but with the heuristic filtering turned on.
3. Less important is understanding how crucial/brittle this method is to the kmeans clustering (if the kmeans clustering does indeed end up being important for improved performance), e.g. what happens if you have a larger k, smaller k (perhaps sampling clusters w/ replacement), etc.

Note if heuristic filtering is entirely responsible for the gians here, I think the paper is still interesting (and even more effective because we don't need this somewhat brittle kmeans step) - but it'd be important to disentangle the performance here.

## Secondary: Lack of numerical rigor in some table results

In general, there are some naked numbers presented in tables that could really use some error bars/additional contextualization, especially since the differences between methods are marginal at best, yet the authors make fairly broad conclusions about the relative efficacy of methods. For example, the numbers in Table 1, especially for Random-Q-CoT, are screaming for error bars. For MultiArith, is 86.2 and 82.8 a meaningful difference? How much variance in performance do you get as you sample different random questions for Random-Q-CoT? It seems like the retrieved demonstration questions, and generated rationales, for Retrieval-Q-CoT are deterministic, if I'm not mistaken. But I'm still curious in general about how much variance there is in these results when you sample different. E.g. if you set temperature != 1, how much does performance vary? For GSM8K in Table 1, a difference of 1 percentage point over Manual-CoT seems rather small and is very close to Random Q-CoT (which again presumably has plenty of variance in the number achievable). So is retrieval not doing better than random in this setting? AQuA results seem more convincing.

## Secondary: 3.1 and 3.2 analysis slightly strange

I wrote some additional thoughts about Section 3, but note this is not as essential to my final recommendation:

I think the essential (interesting!) question that this section aims to answer is: "For hard questions, does Q-Retrieval-CoT underperform because it retrieves similarly hard questions that the model can't do well zero-shot?"

I find the analysis in 3.1 and 3.2 interesting but I feel like they don't quite answer this question as directly as they could.

First, in 3.1, the conclusion that *misleading by similarity* is the blanket reason why Retrieval-Q-CoT underperforms across the board feels premature. The example in Table 2 is convincing. But does this mean the authors are claiming that *every* unresolved example in Figure 2 is because all similar retrieved questions have lexical overlap, with phrases (e.g. *the rest*) that the model is encouraged to consistently misunderstand? This is a very narrow failure case. A more general reason of why Retrieval-Q-CoT might underperform in Figure 2 is just that, for hard questions like those in Figure 2, retrieval might just retrieve similarly hard questions that the model gets wrong. I'm not sure if this is the more specific *misleading by similarity* phenomenon authors describe---if it is, it would be nice to clarify the wording in the paper here.

3.2 is also interesting but strikes me as a rather weird way to answer the  question above. Why was k = 8 decided here? Authors have shown that there exists a cluster with a high zero-shot error rate, but does retrieval indeed tend to retrieve questions within this cluster? More straightforwardly, does Q-Retrieval-CoT tend to retrieve questions that the model gets wrong? In other words, the more straightforward analysis in my opinion is to simply look at the percentage of times the model gets the retrieved demonstration questions correct for any given test question. For harder test questions, perhaps retrieved question accuracy is lower for Retrieval-Q-CoT than Random-Q-CoT, because the retrieved questions are harder. And again, unlike 3.1, this finding would make no claim about any kinds of specific (lexical?) errors propagated from the prompt questions to the test question.

# Minor/questions

- Figure 1 is very helpful for understanding zero shot/manual CoT, but I think the the methods would be much clearer if there were similar visual depictions of how ranadom-q-cot and retrieval-q-cot operate in this figure. The process is not trivial (my understanding: for a test question, retrieve similar questions, generate a rationale via zero-shot CoT, add rationales to prompt, then do standard CoT with the test question)
- Figure 5 would be far more informative if less plots were shown (I don't think you need to visualize the clustering for every single dataset), and instead auhtors focused on a few examples as well as gave concrete examples of what different clusters look like (e.g. in MultiArith, maybe one cluster refers to addition problems, one refers to multiplication, etc.)

# Missing References

This is a matter of principle, but I believe that [Nye et al., 2021](https://arxiv.org/abs/2112.00114) should be cited with any chain of thought prompting work, as it introduced the basic idea prior to it being rebranded "chain of thought"

**Summary Of The Paper:**

This paper proposes a method for automatically constructing chain-of-thought prompts for multistep reasoning in large language models by using zero-shot chain-of-thought prompting (i.e. prepending "let's think step by step" to the generation) to generate rationales for selected prompt questions, and various heuristics for ensuring diversity and simplicity of the generated questions and rationales, including: only keeping fairly short rationales, selecting diverse prompting questions by partitioning the space of questions via kmeans, etc.

The authors first conduct a detailed analysis of what questions might serve as useful prompts for a specific test question. They compare a retrieval-based method (where similar questions are retrieved from a training dataset) with a random method (where random questions), with rationales generated. They observe that the retrieval-based method actually performs worse, and suggest various reasons why this might be the case (e.g., retrieval encourages selecting hard questions, where errors in demonstrations/answers are propagated to the test question). This motivates them to present their final method, Auto-CoT, which takes the lessons learned from the previous section (namely, that diversity matters) and selects diverse questions with fairly straightforward reasoning steps. This method matches and/or outperforms manually constructed chains of thought across ten datasets from arithmetic to commonsense reasoning and symbolic manipulation tasks. Finally, the authors explore a streaming setting, where questions arrive iteratively. These are interesting results though very preliminary.

**Summary Of The Review:**

Overall I think this is an interesting paper with decent results, but I think the strongest possible version of the paper would give us a clear sense of what specifically in the final Auto-CoT model causes increased performance over the simpler variants (Retrieval/Random-Q-CoT). I'm looking forward to the author response and the other reviews and am open to increasing my score.

---

> ### Author Response · Authors · 2022-11-16
> **Response to Reviewer G2Cc**
>
> Thank you for the positive assessment, insightful comments, and being specific in questions for increasing your score.
>
> **1. Understanding why Auto CoT works.**
>
> > "How well does Random-Q-CoT do in Table 3?"
>
> Random-Q-CoT is consistently inferior to Auto-CoT on all the datasets. We have added the results of Random-Q-CoT in Table 3. The results show that Auto-CoT performs significantly better than Random-Q-CoT in all ten datasets at the significance level $p < 0.05$ (or even $p < 0.01 $ for $6$ out of $10$).
>
> > "How much are the simple heuristics buying you?"
>
> To investigate the effect of clustering and simple heuristics, we conduct a further ablation study by removing either one of the design components. As results presented in Table 4 (```w/o clustering``` is implemented based on Random-Q-CoT with simple heuristics), we find that both design components are effective and essential. We have added the analysis in Section 5.3, along with more comprehensive results in Appendix C.2.
>
> As discussed in Section 3, the key challenge of Auto-CoT lies in automatically constructing demonstrations with good questions and reasoning chains. This challenge can be addressed effectively by the two design components: (i) clustering helps obtain the most representative (diverse) questions; (ii) simple heuristics reduce the risk of sampling wrong reasoning chains (statistics presented in Appendix C.3).
>
> > "How crucial/brittle this method is to the kmeans clustering (e.g. what happens if you have a larger k, smaller k)"
>
> To investigate if Auto-CoT is robust against k-means clustering, we set $k$ in different numbers to obtain different numbers of demonstrations. The results are illustrated in Figure 14. We observe that the improvement of Auto-CoT over Random-Q-CoT remains robust against the varying values of $k$. We have added this analysis in Appendix C.5.
>
> **2. Numerical rigor in some table results.**
>
> Thanks for the insightful comments. We have added the mean and standard deviations in Table 1, Table 3, and Table 10. For Random-Q-CoT, we run the experiments with three different random seeds. For Retrieval-Q-CoT, we run the experiments by setting ```temperature=0.7``` to three times. For Auto-CoT, there may be slight randomness in question clustering (e.g., different orders of demonstration), so we also run the experiments with three different random seeds. We have also added those experimental details in Appendix B.2.
>
> **3. About 3.1 and 3.2 analysis.**
>
> > "I'm not sure if this is the more specific misleading by similarity phenomenon authors describe"
>
> Yes. Thanks for your insightful consideration. The reason is exactly as what you described---"for hard questions like those in Figure 2, retrieval might just retrieve similarly hard questions that the model gets wrong". Lexical overlap is just one narrow observation to intuitively explain the similar mistakes derived from the retrieved hard questions. We have clarified the wording in Section 3.1 to make it more accurate.
>
> > "Why was $k = 8$ decided here?"
>
> We basically follow the default setting of 8 demonstrations in Manual-CoT. To show if the observation still holds, we also vary $k$ among \{2, 4, 6, 8\} as shown in Figure 8 (Appendix A.2). We find that the phenomenon is generic when $k$ is different.
>
> > "For harder test questions, perhaps retrieved question accuracy is lower for Retrieval-Q-CoT than Random-Q-CoT because the retrieved questions are harder"
>
> Yes. We calculate the percentage of times the model gets the retrieved demonstration questions correct for hard test questions (dubbed as "Demo Accuracy"). The results verify that Retrieval-Q-CoT indeed fails at the hard test questions compared with Random-Q-CoT.
>
> | Method | Demo Accuracy |
> | --- | --- |
> | Retrieval-Q-CoT | 74.8 |
> | Random-Q-CoT |  87.8 |
>
> **4. Minor/questions.**
>
> > "Visual depictions of how Random-Q-CoT and Retrieval-Q-CoT operate"
>
> Yes. We have added the depictions in Figures 9-10.
>
> > "Figure 5: concrete examples of what different clusters look like."
>
> Yes. We have added concrete examples in Figure 12.
>
> **5. Reference of Nye et al., 2021.**
>
> Yes. We have added the reference in the first paragraph of Section 1.

---

> > ### Comment · Reviewer_G2Cc · 2022-11-19
> > **Thanks for response**
> >
> > Thank you to authors for the detailed response and edits which have addressed a majority of my concerns. After reading the other reviews and the author responses I will increase my score.

---

### Official Review · Reviewer_ajVM · 2022-10-25

**Confidence:** 5
**Correctness:** 4
**Technical Novelty And Significance:** 3
**Empirical Novelty And Significance:** 3
**Recommendation:** 6

**Clarity, Quality, Novelty And Reproducibility:**

Clarity
- The paper is well written and easy to follow. The main paper and the appendix are well-organized and things are written in the right granularity of details.

Novelty
- The proposed method is novel, and combines the best of Zero-shot CoT(kojima et al) and manual CoT(wei et al). It is a great first step to automatically construct CoT prompts based on simple objectives (diversity).


Reproducibility

- All the details required to implement this paper is clearly presented. The authors are also committed to release the code.
- The proposed method is only evaluated on GPT-3, and it would be interesting to see if the proposed method also works on smaller language models.



**Strength And Weaknesses:**

Strengths

- The proposed method is simple yet effective. It is intuitive to implement the proposed method on different tasks.
- The analysis of failure cases of Retrieval-Q-CoT is helpful in understanding the proposed method.

Weaknesses

- Would it be possible to get the best of both worlds by combining Retrieval-Q-CoT and Random-Q-CoT? One can sample tasks similar to the target task, while also sampling questions from other clusters.
- During question sampling the algorithm also has access to answers, and the chain of thoughts. It would be interesting to see if clustering with CoT / Q-A pairs improves the results.


**Summary Of The Paper:**

This paper proposes a method for automatically constructing demonstrations for prompting large language models to perform complex reasoning. The proposed method consists of two main steps. First, partition questions of a given dataset into a few clusters. Second, select a representative question from each cluster and generate its reasoning chain using Zero-Shot-CoT with simple heuristics. The proposed method is evaluated on ten benchmark reasoning tasks from arithmetic reasoning, commonsense reasoning, and symbolic reasoning. Experimental results show that the proposed method performs competitively compared to Manual-CoT which requires manual designs.


**Summary Of The Review:**

This paper proposes a method for automatically constructing demonstrations for prompting large language models to perform complex reasoning. The proposed method is simple yet effective. It is intuitive to implement the proposed method on different tasks. I would recommend that the authors try their method on smaller models and see if the results still hold. The analysis of failure cases of Retrieval-Q-CoT is helpful in understanding the proposed method. I would recommend that the authors try to combine Retrieval-Q-CoT and Random-Q-CoT to get the best of both worlds, or maybe the proposed demo sampling scheme is strictly better than retrieval? Open to increasing my score if all the raised questions get answered in author's response.

---

> ### Author Response · Authors · 2022-11-16
> **Response to Reviewer ajVM**
>
> Thank you for the positive assessment, insightful comments, and being specific in questions for increasing your score.
>
> **1. Combining Retrieval-Q-CoT and Random-Q-CoT.**
>
> To combine Retrieval-Q-CoT and Random-Q-CoT, we investigate a new method dubbed as ```Cluster-Retrieval-CoT```. In detail, we first cluster the questions into $k$ clusters. Then, for each cluster, we retrieve the most similar question to the test question.
>
> The experimental results are added in Table 15. With more diversity involved, Cluster-Retrieval-CoT performs better than Retrieval-Q-CoT. However, the performance is still inferior to Auto-CoT, which uses the representative (e.g., each cluster center) questions from each cluster. In other words, Auto-CoT is better than all our investigated retrieval-related methods. This experiment provides further evidence that using representative skills is helpful (more evidence in Table 12). If we took each demonstration as a kind of skill, representative demonstrations from diverse clusters could cover more alternative skills for solving target questions. We have added the discussion in Appendix D.2.
>
> **2. If clustering with CoT / QA pairs improves the results.**
>
> Yes. Clustering with QA pairs slightly improves the results based on our further experiments (Table 16). However, the computation cost depends on the size of questions because it needs to generate rationales for all questions. As there are often thousands of questions in our investigated tasks, generating the reasoning chains for those questions requires thousands of inferences. Given that API access is often hidden behind the paywalls of LLM providers, there will be much more inference costs. In contrast, our method only needs to generate the reasoning chain for the $k$ (e.g., 8) sampled questions, which enjoys much more cost efficiency. Therefore, we choose the questions only method in our work. We have added the discussion in Appendix D.3.
>
> **3. Try Auto-CoT on smaller models and see if the results still hold.**
>
> Thanks for the suggestion. Indeed, It is interesting to prompt smaller models to be CoT reasoners. As CoT has shown to be an emergent ability of LLMs so far [1], we have not found suitable smaller public models to achieve decent Zero-Shot-CoT results.
>
> As a replacement, we implement Auto-CoT on another 175B version of GPT-3 (text-davinci-001). The text-davinci-001 is a fine-tuned version of GPT-3 and it can be seen as an earlier version of text-davinci-002. It has a weaker Zero-Shot-CoT performance than our previously adopted text-davinci-002. We see that Auto-CoT still attains the most competitive performance under text-davinci-001. We have added the discussion in Appendix C.6.
>
> **References:**
>
> [1] Jason Wei, Yi Tay, Rishi Bommasani, Colin Raffel, Barret Zoph, Sebastian Borgeaud, Dani Yogatama, Maarten Bosma, Denny Zhou, Donald Metzler, Ed H. Chi, Tatsunori Hashimoto, Oriol Vinyals, Percy Liang, Jeff Dean, and William Fedus. Emergent abilities of large language models. Transactions on Machine Learning Research, 2022.

---

> ### Author Response · Authors · 2022-11-29
> **Response to Reviewer ajVM**
>
> Dear Reviewer,
>
> We provided our response two weeks ago. Does it address your questions? We are more than happy to answer any further questions.
>
> Thanks!

---

> ### Author Response · Authors · 2022-12-04
> **Kind Reminder on Your Questions and Our Response**
>
> Dear Reviewer,
>
> Thanks again for your efforts in reviewing our paper, especially indicating "open to increasing my score if all the raised questions get answered in author's response".
>
> We have addressed all your questions in detail. Would you mind checking our response, and confirming if you have further questions?
>
> Best,
> Authors

---

### Official Review · Reviewer_F79B · 2022-10-26

**Confidence:** 5
**Correctness:** 3
**Technical Novelty And Significance:** 3
**Empirical Novelty And Significance:** Not applicable
**Recommendation:** 8

**Clarity, Quality, Novelty And Reproducibility:**

The paper is well-organized and easy to follow. The paper proposes a new automatic method to select demonstration examples. The main concern, from my point of view, is that I am not sure if there is enough modeling novelty.


**Strength And Weaknesses:**

**Strengths**

**1. The tackled problems and proposed solutions are attractive to the community.**

The paper systematically analyzes and compares the strengths and weaknesses of different chain-of-thought prompting methods: including zero-shot-CoT, few-shot-CoT, manual-CoT, and Auto-CoT. The paper conducts case study experiments to look into the limitations of existing methods and proposes improvement directions. Finally, the paper proposes an improved method for Auto-CoT that could achieve a competitive advantage over Manual-CoT.

**2. The experiment part is very detailed and comprehensive.**

**3. The paper is well-organized.**

The writing is good and most of the content is very clear to me.

**Weaknesses/Feedback**

**1. The writing could be improved.**

It would be helpful to draw a table to compare different CoT prompting methods across different dimensions.

How and why shall we make an assumption that “questions of all the wrong demonstrations fall into the same frequent-error cluster”?

Is the selection criteria in section 4.2 reasonable? Namely, why do we not choose questions with more than 60 tokens and rationales with more than 5 reasoning steps?


**2. Some experimental details are missing.**

The experimental details for Table 1 are not very clear and lack some details. For example, how the manual-CoT examples are built. What is the number of demonstration examples?

The experimental details for the Codex baseline are missing. I am curious about the instructions you used to prompt the Codex model.


**3. It would be better to discuss more recent related work.**

I understand some work has been released very recently. Since this work is closely related to the paper, it would be nice to include it in the revised paper. Recent work includes but is not limited to:
[1] Calibrate Before Use: Improving Few-Shot Performance of Language Models, 2021
[2] Dynamic Prompt Learning via Policy Gradient for Semi-structured Mathematical Reasoning, 2022
[3] Complexity-Based Prompting for Multi-Step Reasoning, 2022
[4] Learn to Explain: Multimodal Reasoning via Thought Chains for Science Question Answering, 2022
[5] What Makes Pre-trained Language Models Better Zero/Few-shot Learners?, 2022

**4. Is it possible to apply Auto-CoT to sample the examples with manually designed CoT?**


**Summary Of The Paper:**

The paper compares different chain-of-thought (CoT) prompting methods proposed recently and shows their limitations via case studies. The paper further proposes a new automatic chain-of-thought prompting method, Auto-CoT. Auto-CoT samples questions from representative question clusters and generates reasoning chains to construct demonstrations. Results on a wide range of datasets show that it performs competitively compared to Manual-CoT, which requires manual designs.

**Summary Of The Review:**

The paper is overall high quality. The proposed method is beneficial to the community in developing more performant few-shot GPT-3 models. The experiments are comprehensive and the proposed model achieves competitive results on various datasets.

---

> ### Author Response · Authors · 2022-11-16
> **Response to Reviewer F79B**
>
> Thanks for your positive assessment and constructive feedback.
>
> **1. About writing.**
>
> > "It would be helpful to draw a table to compare different CoT prompting methods across different dimensions."
>
> Yes, we have added the comparison of different CoT prompting methods across different dimensions in Table 8.
>
> > "How and why shall we make an assumption that questions of all the wrong demonstrations fall into the same frequent-error cluster?"
>
> This phenomenon essentially motivates our solution. Firstly, we find that Retrieval-Q-CoT often fails like Zero-Shot-CoT (Section 3.1). Then, we are interested in investigating the reason for the phenomenon. We find that Retrieval-Q-CoT tends to retrieve demonstration questions with similar errors---such errors are often semantically similar and thus fall into the same cluster (Section 3.2). Finally, we are inspired to devise a cluster-based technique to reduce the risk of retrieving multiple similar questions inside the frequent-error cluster (Section 3.3).
>
> >"Is the selection criteria in section 4.2 reasonable? Namely, why do we not choose questions with more than 60 tokens and rationales with more than 5 reasoning steps?"
>
> The selection follows the criteria in Manual-CoT [1]. On the one hand, we hope to make the demonstrations in similar forms to those in Manual-CoT for a fair comparison. On the other hand, the selection helps reduce the proportion of wrong demonstrations (statistics in Figure 13). We observe that the selection criteria improve the model performance, as verified in our new experiments (Table 10).
>
> **2. About experimental details.**
>
> >  "The experimental details for Table 1 are not very clear and lack some details. For example, how the manual-CoT examples are built. What is the number of demonstration examples?"
>
> The examples are from the original Manual-CoT paper [1]. There are 8 examples for MultiArith and GSM8K, and 4 examples for AQuA. We have added the details in Footnote 1 on Page 3.
>
> >  "The experimental details for the Codex baseline are missing. I am curious about the instructions you used to prompt the Codex model."
>
> The instructions are the same as those for GPT-3 (their formats are illustrated in Figure 4). Those instructions work well for the Codex model. The only exception is that we set an extra argument ```stop=["Q:"]``` (as the end-of-sequence token in generation) when invoking the OpenAI API to avoid generating additional questions after answering the test question. We have added the detail in Appendix B.2.
>
> **3. Discussion about more recent related work.**
>
> Thanks for pointing out those related papers. We have added the discussions with those studies in Section 2.
>
> **4. Apply Auto-CoT to sample the examples with manually designed CoT.**
>
> Great point. It is possible to apply Auto-CoT to sample the examples with manually designed CoT. We perform further analysis on the GSM8K and AQuA datasets which have training sets with manually designed reasoning chains. We cluster the questions from the training set and use the gold reasoning chains as our demonstrations. We observe that Auto-CoT can be further enhanced once manually designed CoT is available. We have added the discussion in Appendix D.1.
>
> **References:**
>
> [1] Jason Wei, Xuezhi Wang, Dale Schuurmans, Maarten Bosma, Ed Chi, Quoc Le, and Denny Zhou. Chain of thought prompting elicits reasoning in large language models. In Thirty-sixth Conference on Neural Information Processing Systems (NeurIPS), 2022.

---

### Author Response · Authors · 2022-11-16
**General Response**

We want to thank all the reviewers for the positive assessment, insightful comments, and being specific in questions for increasing their scores. We have revised our paper accordingly and provided individual responses to each reviewer. The main changes to the paper are summarized as follows:

1. Improving the presentation by (i) adding a comparison of different CoT prompting methods across different dimensions (Table 8); (ii) adding illustrations of Random-Q-CoT and Retrieval-Q-CoT in Figures 9--10; (iii) adding concrete example questions from different clusters (Figure 12); (iv) adding the discussions about more recent work in Section 2; (v) adding more experimental details including the settings in Table 1 and implementation of Codex (Appendix B.2); (vi) updating the results in Table 1, Table 3, and Table 10 with mean and standard deviations.
2. Adding more experimental results and discussions, including (i) an ablation study of clustering and simple heuristics (Appendix C.2); (ii) influence of the number of clusters (Appendix C.5); (iii) applying Auto-CoT to more LLMs (Appendix C.6).
3. Adding discussions about different design choices, including (i) applying Auto-CoT to sample demonstrations with manually designed CoT (Appendix D.1); (ii) combining Retrieval-Q-CoT and Random-Q-CoT (Appendix D.2); (iii) clustering with CoT QA pairs (Appendix D.3); (iv) comparison with other zero-shot CoT prompts (Appendix D.4); (v) one-shot setting by using one demonstration from the cluster where the test sample fits (Appendix D.5).

---

### Decision · Program_Chairs · 2023-01-20

**Decision:**

Accept: poster

**Justification For Why Not Higher Score:**

There are some issues regarding clarity of the writing and the main claim to be too strong.

**Justification For Why Not Lower Score:**

Please see the detailed AC-Reviewer meeting notes.

**Metareview: Summary, Strengths And Weaknesses:**

The paper presents an approach for automatic prompting to elicit chain-of-thought reasoning in large language models without needing manually-designed demonstrations.
All reviewers agreed that the paper addresses an important problem and appreciated the depth of the experiments and analysis. However, some reviewers raised concerns about the lack of support for the claim of automatic chain of thought prompting and the limited improvement provided by the proposed method over manual prompts.
After the discussion among the reviewers, some of these concerns were unresolved. However, after carefully considering the reviews, the meeting between AC and reviewers, and the author response, it does seem like that addressing the second concern is not really the focus of this paper, as manual prompts can be considered as a kind of an upperbound for performance. The first major concern can be addressed by asking the authors to rephrase the "automatic CoT prompting" claim to make things more clear. It does seem like the proposed approach is an automatic "example selection" for zero-shot prompting (Kojima et al., 2022).


**Note From Pc:**

if the above contains the word "oral" or "spotlight" please see: "oral" presentation means -> notable-top-5% and "spotlight" means -> notable-top-25%. As stated in our emails, we are disassociating presentation type from AC recommendations

**Summary Of Ac-Reviewer Meeting:**

Unfortunately only 2 of the reviewers attended the meeting. Here are the meeting notes:
1- The automatic chain of thought (CoT) prompting claim is not very well supported.
The approach basically does example selection instead of automatic CoT prompting, which is already studied in similar papers.
2- In addition, the results do not provide significant improvements over manual prompts.
Another concern is the cost of the approach, if the data size increases, the cost of automatically selecting prompts would likely increase.
The reviewers also think that if the method of prompt selection was more novel, it would have made the paper more interesting.
The style of prompts is similar to the one used by Kojima et al, 2022.

However, I pushed for comment from other reviewers who were supportive of the paper.
One of them pushed back about the above comments:

```
Overall I want to push back on most of the concerns here. Probably the one that I can most empathize with is the claim that it's not really "automatic CoT prompting", it's example selection, so if there is related literature on example selection this somewhat decreases my perception that the paper should be accepted.
If it were up to me alone, I would still accept the paper, as I think it's a useful contribution, but I'll defer the final decision to the AC.
1- The automatic chain of thought (CoT) prompting claim is not very well supported.
The approach basically does example selection instead of automatic CoT prompting, which is already studied in similar papers.
2- In addition, the results do not provide significant improvements over manual prompts.
Another concern is the cost of the approach, if the data size increases, the cost of automatically selecting prompts would likely increase.
The reviewers also think that if the method of prompt selection was more novel, it would have made the paper more interesting.
The style of prompts is similar to the one used by Kojima et al, 2022.
- I don't understand this concern. The goal should not be able to improve over manual prompts. We can think of manual prompts as an upper bound, gold-standard - i.e. getting humans to exactly annotate how to do tasks. The fact that we can even come close (and in some cases outperform) manual human prompts without having to actually get human prompts seems like the success of this approach in my books.
- The important comparison is whether smartly selecting examples for CoT prompting is better than randomly selecting (i.e. Auto CoT vs Random-Q-CoT). I think the improvements in this setting are sufficient (again not huge, but sufficient), and the ablations I asked for solidify my intuition that we're actually doing something that works here.
- Let me know if you disagree with how we should be interpreting the results, in particular comparisons to manual and random CoT.
- I'm not concerned about this - the real upper blocker on cost is getting manual prompts which takes human annotators and $$$ (again the gold standard upper bound is manual CoT). Isn't the cost here just the computational/data storage cost of doing nearest neighbor search in a larger space of examples? This seems negligible to me - let me know if I've misunderstood.
```


After carefully reading all the reviews, author response, and confidential note to AC, and the review notes, I would vote for **Accepting** the paper and I don't view the critics of the negative reviewer as major concerns. That being said, I believe the authors could be asked to make the writing more clear and potentially not use the "automatic CoT prompting" phrase.